# Real-time ultrasound evaluation of CORE muscle activity in a simultaneous contraction in subjects with non-specific low back pain and without low-back pain. Protocol of an observational case-control study

María Cervera-Cano[1,2], María Carmen Sáez-García[1], David Valcárcel-Linares[1], Samuel Fernández-Carnero[1]*, Luis López-González[1,3], Tomás Gallego-Izquierdo[1], Daniel Pecos-Martin[1]

1 Facultad de Medicina y Ciencias de la Salud, Departamento de Enfermería y Fisioterapia, Grupo de Investigación en Fisioterapia y Dolor, Universidad de Alcalá, Alcalá de Henares, Spain, 2 Department of clinical trials, University Hospital 12 de Octubre, Madrid, Spain, 3 Department of physiotherapy, Ramón y Cajal Hospital, Madrid, Spain

* samuelfernandezcarnero@gmail.com

## Abstract

Non-specific low back pain represents 90–95% of all cases of low back pain and it has a prevalence of 18% in the adult population, assuming a great socioeconomic impact. The main objective of this observational case-control study study is to evaluate if there are differences in the simultaneous contraction of the core muscles between nonspecific low back pain and healthy subjects. This study will be carried out in the Physiotherapy department of the University of Alcalá. Eighty-two participants <18 years old, will be recruited, paired with NSLBP (n = 41) and healthy (n = 41). The main outcome will be the onset muscle contraction of lateral abdominal wall (internal oblique, external oblique and transversus abdominis), pelvic floor, lumbar multifidus and respiratory diafragm. The maneuvers that the subjects will perform will be abdominal drawing in maneouver, contralateral arm lift, valsalva, and voluntary pelvic floor contraction in sitting and standing. As a secondary objective, to analyze the amount of contraction of each muscle group and the capacity of the diaphragms to be excreted in both groups of subjects. Finally, to relate pain and disability.

## Introduction

Low back pain (LBP) has been the leading cause of disability since 1990 [1]. In 85–90% of cases, the exact cause of the pain cannot be determined with certainty and patients are classified as Non-Specific Low Back Pain (NSLBP) [1]. Pain has a prevalence of 18% in the adult population, which has a major socioeconomic impact worldwide [2]. In 10% of people suffering from LBP it becomes chronic [1], as it lasts longer than twelve weeks in a year [2]. Acute LBP has been defined by other guidelines as less than 4,6 or 12 weeks duration [2]. The type of work, obesity and unfavorable habits in daily life have also been considered risk factors that

**Data Availability Statement:** Deidentified research data will be made publicly available when the study is completed and published.

**Funding:** The authors received no specific funding for this work. The funders had and will not have a role in study design, data collection and analysis, decision to publish, or preparation of the manuscript.

**Competing interests:** The authors have declared that no competing interests exist.

increase the likelihood of NSLBP [2]. It has been proposed that maladaptive social and psychological factors (depression, anxiety, catastrophism and low self-efficacy) may play an important role in the persistence of pain [3]. However, the role of musculoskeletal factors remains unclear.

There is no consensus on an exact definition of the core, this term refers to the abdomino-pelvic functional unit that involves not only vertebral segments with their corresponding passive structures that support it or the neural system, but also the four fundamental pillars that conform it: Lumbar multifidus (LM), Lateral Abdominal Wall (LAW), Respiratory Diaphragm (DPH) and pelvic floor (PF) [4, 5]. All this musculature forms a cylinder that works in synergy, producing a crossroads of lumbo-pelvic forces that provide trunk stability, better trunk control, efficient movement, good balance, and coordination, as well as better motor control (postural firmness and alignment). However, although the term core refers to a number of muscle groups, it should be considered a functional rather than an anatomical term [4, 5].

Various methods have been used to assess muscle behavior and/or core morphology with electromyography (EMG) being the 'Gold standard'. Increasingly, the use of RUSI ('Rehabilitative ultrasound imaging') and real-time ultrasound (RTUS) has become an alternative to Magnetic Resonance Imaging (MRI) as it has been shown to be reliable in comparison with the 'gold standard' (EMG or MRI) [6]. The use of Motion Mode (M-Mode) in ultrasound (US) has been proven to be a useful tool for assessing muscle contraction in people with NSLBP [4, 7]. Perhaps one of the limitations in performing US studies of the functional unit of the core simultaneously has been the lack of a measurement tool that supports several probes at the same time.

Numerous authors have attempted to explain the causes of NSLBP of musculoskeletal origin related to alterations in movement, muscle activity and/or changes in muscle atrophy in patients with NSLBP compared to asymptomatic participants [8]. These changes appear to be important factors in the evaluation and treatment of patients with LBP [6].

In relation to the definition of core and motor control, it could be understood that not only the morphological situation of the musculature and its contraction in isolation seems to be important, but also the coordination between the muscle groups that conform this functional unit [9]. Therefore, studies that analyze the global muscular behavior of the core and its morphology in a situation of pain in comparison with healthy subjects would give much information about the possible differences existing between both groups of subjects and their possible implication in NSLBP.

In the present protocol, the synergies between the muscle groups of the core that could be involved in NSLBP pain measured with US will be observed as the main objective. It is considered that an alteration in the synergy and in the global behavior of this functional unit may be related to the perpetuation or cause of NSLBP. As a secondary objective, to analyze the amount of contraction of each muscle group and the capacity of the diaphragms to be excreted in both groups of subjects. Finally, to relate pain and disability.

## Materials and methods

### Study design

An observational case-control study is proposed. The study research will be developed according to the Strengthening the Reporting of Observational Studies in Epidemiology (STROBE) [10] and the Guidelines for Reporting Reliability and Agreement Studies (GRRAS) [11]. Data will be collected between December 2021 and February 2023. The study was conducted according to the guidelines of the Declaration of Helsinki and approved by the Institutional Review Board (or Ethics Committee) of Ethical Committee for Research and Animal Experimentation (CEIM) of the University of Alcalá (CEIM/HU/2019/41).

## Participants

Participants will be chronic NSLBP subjects, and asymptomatic, according to the selection criteria and recruited through non-probability convenience sampling. Participants will contact the University of Alcalá, in Madrid. The participant will be provided with the "Adult Research Patient Information Sheet" (S1 Appendix), an informed consent (S2 Appendix), a 'Participant Data Collection Form' (S3 Appendix) and the Low Back Pain Questionnaire (S4 Appendix) designed by the research team. Once the subject has signed the Informed Consent form and completed the Data collection form and the Low Back Pain Questionnaire, the inclusion and exclusion criteria will be reviewed by a sub-investigator (D.V.L or L.L.G). Finally, the participant will be assigned to the case or control group.

## Selection criteria

The inclusion criteria for the chronic NSLBP group will be:

- Subjects who suffer from chronic NSLBP according to the criteria defined in the scientific literature [1, 2].

- Participants aged between 18 and 60 years old [12].

- To have a history of NSLBP at least twelve weeks in the last year [2]

- Have scored their pain greater than or equal to 3 points out of 10 on a Visual Analog Scale (VAS) [13, 14].

The control group will be subjects asymptomatic, who have neither acute nor chronic low back pain at the time of the measurement, nor who have suffered it in the last year since the day of the visit.

Surgical intervention, vertebral pathology (fractures, cancer, or infection) inability to be in the position or to perform the selected maneuvers will be the criteria for exclusion from this study. Pregnancy will also be a criterion for exclusion [12].

## Sample size calculation

The sample size calculation has been carried out with the program 'GPower 3.1.5' and based on the main inter-subject factor of a mixed 4-by-2 analysis of variance (ANOVA). An effect size of 0.25 was estimated, with a correlation between repeated measurements of 0.50, a power of 80% and a level of 0.05. The resulting sample size was 82 subjects.

## Outcomes

The primary outcome will be the onset muscular contraction of the four core muscle groups during the Abdominal Drawing in Maneuver (ADIM), Contralateral Arm Lift (CAL), Valsalva maneuver, and voluntary contraction of the PF, in sitting and standing positions:

a.  Onset muscular contraction of the three muscles that conform the Lateral Abdominal Wall (LAW); Transversus abdominis (TrA), Internal oblique (IO), External oblique (EO).

b.  Onset muscular contraction of Lumbar Multifidus (LM).

c.  Onset excursion of the Diaphragm (DPH).

d.  Onset excursion of the Pelvic Floor (PF).

The secondary outcomes are; the percentage of change in muscle thickness in the LM and LAW (IO,EO,TrA) represented as average contraction thickness (cm)–average thickness at

rest (cm) / average thickness at rest (cm)*100 [12]; the excursion of the DPH [15] and the excursion of the bladder trough the contraction of the PF muscles [16]. Finally, the degree of disability through the Oswestry disability questionnaire) [17].

## Procedures

The data will be collected by an expert physiotherapist in RUSI with 10 years of experience (S. F.C) and the principal investigator (M.C.C) with 60 hours of training in US [18]. Both researchers (S.F.C and M.C.C) will be blinded to group assignment during data collection. Two sub-investigators (D.V.L and L.L.G) will collect the participant's Data collection Form and assign him/her to the case or control group. Once the participant has completed all the information and has been assigned to the corresponding group, it will be filed in the investigator file to avoid breaking the blind by principal investigator.

Four wireless US probes, two linear with a 7,5–10 MHz bandwidth and two convex with a 3,5–5 MHz bandwidth will be used. These probes will be held in place by a customized fixation belt manufactured by the research team that has been resulted as a utility model filed with the Spanish patent and trademark office (OEPM; n° 202131486) (Fig 1). The customized fixation belt will allow the simultaneous collection of ultrasound images and videos.

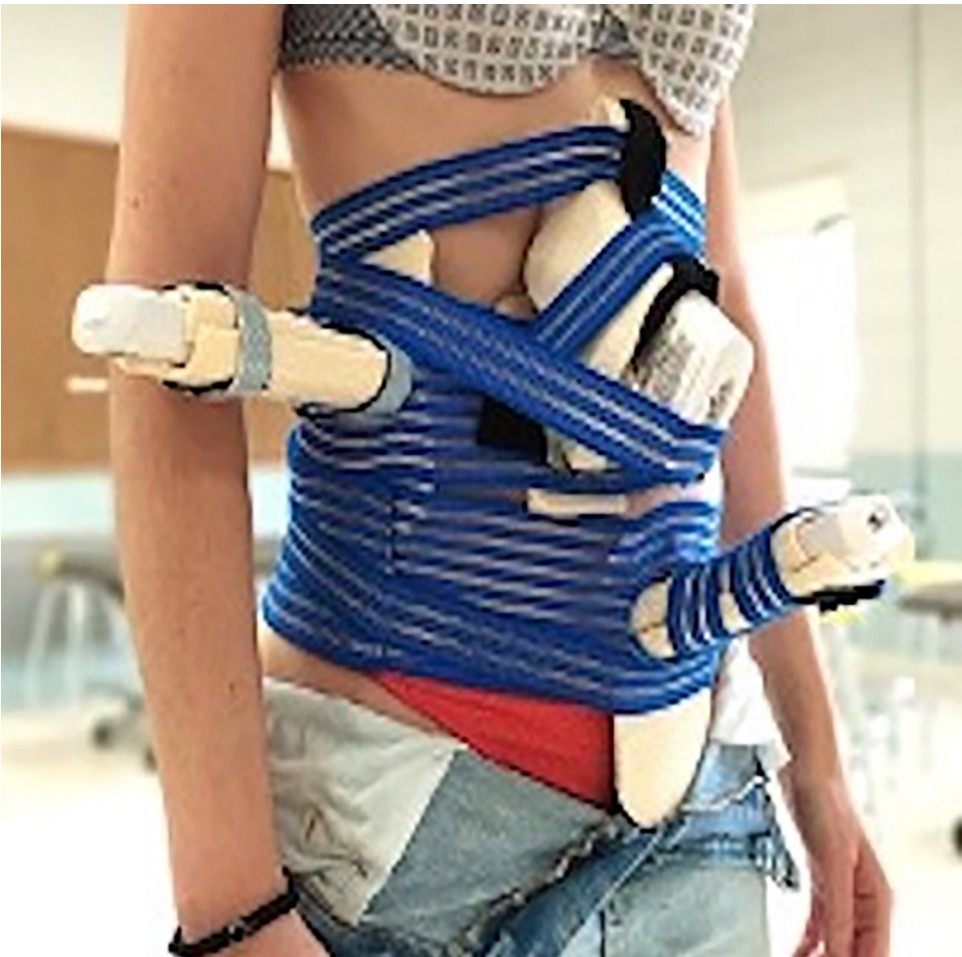

**Fig 1. Customized fixation belt.**

Three of the probes corresponding to the LAW, LM and DPH will be positioned on the right side of the patient as the sound amplifying window offered by the liver is needed to measure the excursion of the DPH [15]. The fourth probe corresponding to the PF is placed transversely and suprapubically on the subject [16]. The probes placement will correspond to the measurement areas of each muscle group that have been previously validated by different authors [15, 16, 19, 20]. The four US images will be visualized and unified in a computer through a video capture card, where the onset muscle contraction and the morphology of the muscles will be analyzed.

An US analysis will be carried out to evaluate the onset muscle contraction of the four core muscle groups in a global and synergic manner, as well as their individual morphological characteristics during the ADIM, CAL, voluntary PF contraction and Valsalva maneuvers in sitting and standing. The measurements will be taken in these two positions since they have been assumed in as possible risk factors for the appearance of NSLBP and its maintenance [21].

The maneuvers and the positions will be randomized by means of opaque envelopes that will be assigned to each patient to eliminate possible biases. All tests will be performed once in sitting and standing positions. with a 2 minutes break between maneuvers. The authors of this study considered that 2 minutes break between each maneuver and 2 minutes of rest between positions, being low intensity tests, could be enough to avoid participant fatigue. All the maneuvers will be taught before each measurement.

**Abdominal Drawing in Maneuver (ADIM).**   The ADIM is designed to facilitate coactivation of the TrA and LM to stabilize the trunk prior to limb movement [22]. Subjects will receive a traditional training with tactile and verbal instructions in sitting and standing positions before the measurement [22]. To perform ADIM, the participant will be taught to "Take a relaxed breath in and out, hold the breath out, and then draw in the lower abdomen without moving their spine" [20].

**Contralateral Arm Lift (CAL).**   The CAL maneuver attempts to identify the muscular activity of the LM and its size during the execution of the test. With the elbow fully extended and the wrist in a neutral position, the participant will raise the arm to shoulder height. The subject will hold the position for 2 seconds [23].

**Voluntary pelvic floor contraction.**   To perform the voluntary PF contraction, the anterior superior iliac spine and the posterior superior iliac spine will be aligned horizontally to avoid the displacement of the base of the bladder. It will be allowed the contraction of the TrA as a synergic muscle. Once the optimum measuring position has been reached, a command will inform the participant to activate the PF: "try to close the urethra as if you wanted to hold back urine". The PF contraction will be maintained for 6–10 seconds [24].

**Valsalva maneuver.**   The Valsalva maneuver is a breathing pattern that consists of forced exhalation with the glottis closed to increase the stability of the spine by increasing the intra-abdominal pressure, since the entire abdominal musculature is activated [25]. The sequence given to the subjects to carry out the Valsalva maneuver will be as follows (1)" Take the air into the ears with the mouth and nostrils closed", (2) "apply pressure towards the pelvic floor, as if they intended to defecate". No instruction shall be given on the contraction or relaxation of the abdominal muscles or the PF. The muscular activity will be recorded within 3 seconds of the participant's maximum contraction [25].

The maneuvers and its characteristics are summarized in Table 1.

The methodology of the US evaluations is summarized in Table 2.

## Data processing

The data will be collected from December 2021 to February 2023. Written informed consent will be obtained from all participants prior to recruitment. The participant Data Collection

**Table 1. Maneouvers and its characteristics.**

| Maneuver | Verbal command | Muscle contraction | Rest between maneouvers |
|---|---|---|---|
| **ADIM** [22] | Take a relaxed breath in and out, hold the breath out, and then draw in the lower abdomen without moving their spine" | At the end of the exhalation | 2 minutes |
| **CAL** [23] | Raise the extended arm to shoulder height and hold it for two seconds. Then lower it naturally. | 2 seconds | 2 minutes |
| **Voluntary PF contraction** [24] | try to close the urethra as if you wanted to hold back urine" | 6 seconds -10 seconds | 2 minutes |
| **Valsalva** [25] | "apply pressure towards the pelvic floor, as if they intended to defecate' | 3 seconds | 2 minutes |

ADIM, Abdominal Drawing in Maneuver; CAL, Contralateral Arm Lift; PF, Pelvic Floor

form that contains information as socio-demographic data (height, weight, and personal medical history) will be completed by the participant and a 10-question questionnaire (S4 Appendix) designed by the research team that will allow the participant to be assigned to the case or control group. The Visual Analog Scale (VAS) [14] as well as the Oswestry Dissability Index scale [17] will be completed only by subjects who have been assigned to the case group. To ensure the authenticity of the data, all individuals will complete the questionnaires independently.

Once the participant has completed all the forms, the customized fixation belt will be placed on the subject, with its corresponding US probes. With the US M-Mode, the start of movement of each specific muscle group will be identified. These results will be compared with the morphological characteristics of each muscle group such as the % of change in thickness of the LM and the LAW, and the endopelvic and respiratory diaphragms excursions.

The US videos and images of each probe will be displayed in four tablets through the USG Wireless program. These tablets are connected to a computer through a customized video capture card.

To objectify the onset muscle contraction of each muscle group, the OBS (Open Broadcaster Software® V.28.1.2 Boston, MA 02110–1301 USA) will be used. The OBS will allow to observe the four videos all at once, where the specific and global muscle contraction can be

**Table 2. Methodology of the ultrasound evaluations.**

| Muscle | Type of probe | Transducer placement | Transducer bandwidth | Caliper placing |
|---|---|---|---|---|
| LAW (IO, EO, TrA) [26] | Linear | Right abdominal wall at the midpoint between the lower angle of the thoracic cage and the iliac crest. The medial edge of the transducer shall be placed approximately 10 cm from the alba linear. | 7'5MHz | Distance [cm] between the superficial and deep border of each muscle belly, which is marked by the hyperechoic fascial lines. The fascial lines will not be included in the measurement. |
| LM [27] | Linear | L4-L5 facet joint over the articular pillar and the belly of the L4-L5 muscle over the articular pillar and the belly of the ML muscle. | 5MHz | Distance (cm) from the most posterior portion of the zygapophyseal joint of L4-L5 and the muscle surface separating it from the subcutaneous cellular tissue |
| PF [24] | Curved | Suprapubic, just in the mid-sagittal and transabdominal plane. The probe will be tilted caudally and posteriorly to obtain a clear image of the postero-inferior aspect (base) of the bladder, which will vary depending on the bladder fullness of the participants. | 2MHz-5MHz | Measure the distance (cm) from the base of the bladder to the central part of the convex probe. |
| DPH [15] | Curved | Right anterior axillary line, Subcostal. The gallbladder and inferior vena cava will be taken as anatomical landmarks. | 2'5MHz -3'5MHz | Distance (cm) from the most distal apex of the right hemidiaphragm to the midpoint of the convex probe coinciding with the hyperechogenic curved line of the diaphragm dome. |

LAW, Lateral Abdominal Wall; LM, Lumbar Multifidus; PF, Pelvic Floor; DPH, Diaphragm.

visualized in a simpler and clearer way. In addition, a stopwatch will be used in the same program to visualize the onset muscle contraction.

The collected data will be saved and imported to a personal laptop for later measurement by a blinded examiner using the program Free software FIJI® V.2.9.0 Cambridge CB3 0HA, UK [28].

## Statistical analysis

The analysis of the assumption of normality will be carried out with the Kolmogorov-Smirnov test with Lilliefors' correction, as the sample is larger than 50 subjects per group. For the descriptive analysis of the continuous quantitative variables the mean and standard deviation (SD) will be used in the case of meeting the normality and the median and first and third quartiles in the opposite case. In the case of nominal variables, absolute and relative percentage frequencies will be reported. The analysis of the homogeneity of the groups in the demographic variables will be carried out with the t-Student test for independent samples in the case of the continuous variables that comply with the assumption of normality and by means of the Mann-Whitney U test in the case of those that do not. For nominal variables, the Pearson chi-square test will be used or the Fisher exact test if the assumptions for the first one is not met.

The Intraclass Correlation Coefficient (ICC) will be used for the analysis of the intra and inter-reliability of the onset muscular contraction and ultrasound measurements, under the assumption of a mixed model of two factors and absolute agreement, using the average of 3 measurements (ICC3,3) [29]. It will be performed in 20 participants of the sample during standing voluntary pelvic floor contraction maneuver. The presence of a systematic error will be assessed by the statistical significance of the analysis of variance (ANOVA) used for the calculation of the CCI. The standard error of measurement (SEM) will be calculated as the square root of the mean square of the ANOVA error used for the calculation of the JRC, and the minimum detectable difference (MDD) at 95% confidence, using the formula $MDD_{95} = SEM * \sqrt{2} * 1.96$. The SEM and MDD will also be reported as a percentage of the sample mean. The Bland-Altman agreement charts will then be produced, estimating the agreement limits as the mean difference $\pm 1.96*DT$ [30].

For the assessment of compliance with the assumption of homoscedasticity, a visual inspection of the graph will be carried out and the relationship between the differences and the mean values, and between the differences with an absolute value and the mean values, will be analyzed using the Pearson correlation coefficient [31].

For the analysis of the differences between the groups in the continuous quantitative dependent variables that meet the assumption of normality, a mixed 4-by-2-by-2 covariance analysis (ANCOVA) will be used, with the maneuver (ADIM, CAL, Valsalva, voluntary PF contraction) and the position (sitting, standing) as intra-subject factors, the group (pain, healthy) as an inter-subject factor, and the BMI (Body Mass Index) as a covariant. The association between abdominal muscle thicknesses with gender, age, and BMI was examined using Pearson's correlation coefficient and multiple regression analysis. Compliance with the assumption of sphericity will be assessed with the Mauchly test and that of equality of covariance matrices with the Box test. The partial eta-square coefficient (p2) will be used as an estimator of the size of the ANCOVA main effects and interactions.

It will be performed the Kruskal-Wallis H test for analyze the differences between independent variables (cases and controls) and the onset muscular contraction, as a qualitative ordinal dependent variable. On the other hand, the t-Student test for independent samples with Bonferroni correction will be used for the analysis of post-hoc pairwise comparisons. Cohen's d ($d = 2t / \sqrt{g}$) will be used as estimator of the effect size of the post hoc comparisons [31].

If any continuous quantitative dependent variable does not meet the assumptions for carrying out the ANCOVAs, robust analogous methods will be used that do not require compliance with those assumptions, following the procedures described by Wilcox [31]

All analyses will be carried out with the software R Version 4.1.0 (R Core Team (2021). R: A language and environment for statistical computing. R Foundation for Statistical Computing, Vienna, Austria. URL https://www.R-project.org/). An α level of 0.05 with 95% confidence intervals (CI) will be assumed for all analyses.

## Supporting information

**S1 Appendix. Patient information sheet (hip).**
(DOCX)

**S2 Appendix. Declaration of informed consent.**
(DOCX)

**S3 Appendix. Participant data collection form.**
(DOCX)

**S4 Appendix. Low Back Pain Questionnaire.**
(DOCX)

**S1 File.**
(PDF)

## Author Contributions

**Conceptualization:** María Cervera-Cano, María Carmen Sáez-García, Samuel Fernández-Carnero, Daniel Pecos-Martin.

**Data curation:** María Cervera-Cano, David Valcárcel-Linares, Luis López-González.

**Formal analysis:** María Cervera-Cano, David Valcárcel-Linares.

**Funding acquisition:** María Cervera-Cano, María Carmen Sáez-García.

**Investigation:** María Cervera-Cano, Luis López-González.

**Methodology:** María Cervera-Cano, María Carmen Sáez-García, Samuel Fernández-Carnero.

**Project administration:** María Cervera-Cano, Tomás Gallego-Izquierdo.

**Resources:** María Cervera-Cano, Daniel Pecos-Martin.

**Software:** María Cervera-Cano, Samuel Fernández-Carnero.

**Supervision:** María Cervera-Cano.

**Validation:** María Cervera-Cano.

**Visualization:** María Cervera-Cano.

**Writing – original draft:** María Cervera-Cano.

**Writing – review & editing:** María Cervera-Cano.

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
