## [Decision Letter · Decision Letter 0]

15 Nov 2022

PONE-D-22-29823Real-time ultrasound evaluation of CORE muscle activity in a simultaneous activation in subjects with non-specific low back pain and without low-back pain. Protocol of an observational cross-sectional study.PLOS ONE

Dear Dr. Fernández Carnero,

Thank you for submitting your manuscript to PLOS ONE. After careful consideration, we feel that it has merit but does not fully meet PLOS ONE’s publication criteria as it currently stands. Therefore, we invite you to submit a revised version of the manuscript that addresses the points raised during the review process.

We look forward to receiving your revised manuscript.

Kind regards,

Mehmet Cudi Tuncer, Ph.D.

Academic Editor

PLOS ONE

Journal Requirements:

2. We note that your submission may include questionnaire items that may have been previously published. The reproduction of previously published work has implications for the copyright that may apply to these publications. We would be grateful if you could clarify whether you have obtained permission from the original copyright holder to republish these items under a CC BY license. If you have not obtained permission to publish these items please remove them from your manuscript. You may wish to replace the text you have removed with relevant question numbers/ brief descriptions of each item; please be sure to include any relevant references and in-text citations.

"The funders had and will not have a role in study design, data collection and analysis, decision to publish, or preparation of the manuscript."

"he authors have declared that no competing interests exist."

6. Please upload a copy of Figure 3, to which you refer in your text on page 7. If the figure is no longer to be included as part of the submission please remove all reference to it within the text.

Reviewers' comments:

Reviewer's Responses to Questions

**Comments to the Author**

1. Does the manuscript provide a valid rationale for the proposed study, with clearly identified and justified research questions?

Reviewer #1: Yes

2. Is the protocol technically sound and planned in a manner that will lead to a meaningful outcome and allow testing the stated hypotheses?

Reviewer #1: Yes

3. Is the methodology feasible and described in sufficient detail to allow the work to be replicable?

Reviewer #1: Yes

4. Have the authors described where all data underlying the findings will be made available when the study is complete?

Reviewer #1: No

5. Is the manuscript presented in an intelligible fashion and written in standard English?

Reviewer #1: Yes

6. Review Comments to the Author

You may also provide optional suggestions and comments to authors that they might find helpful in planning their study.

Reviewer #1: This is a study protocol that investigates the activation time of core muscles via ultrasound in ‎subjects with and without low back pain.‎

Abstract:‎

‎“Observational case-control study." Why did the authors mention “Protocol of an observational ‎cross sectional study” in the title?‎

Method:‎

Who will diagnose NSLBP? Physician or questionnaire?‎

Will the authors consider acute NSLBP or chronic NSLBP?‎

‎“History of LBP at least three full weeks” for example, a person with 25 days of NSLBP is ‎considered to have acute LBP, while “more than three months in the last year” is a definition for ‎chronic NSLBP.‎

Will the authors include patients with a history of recent participation in core exercises in the ‎past six months? ‎

‎“Balance Mass Index (BMI)”? body or balance?‎

Did the authors consider the confounding effect of food consumption on the muscle thickness ‎evaluations? (Eur Spine J. 2011 Aug;20(8):1312-7. The effect of food consumption on the ‎thickness of abdominal muscles, employing ultrasound measurements) (Man Ther. 2015 ‎Feb;20(1):194-9. Reversal time of postprandial changes of the thickness of abdominal muscles ‎employing ultrasound measurements).‎

Will the participants be taught how to perform ADIM before measurements?‎

Where will the authors assess the internal oblique, external oblique, and transverse abdominis?‎

Considering the effect of sex, age, and BMI on muscle thickness, why won't the authors adjust ‎their analysis for sex, age, and BMI? ( J Physiol Anthropol. 2016 Aug 23;35(1):17. ‎Measurement of superficial and deep abdominal muscle thickness: an ultrasonography study)‎

Is the length of the VAS measurement tool 10 cm? The image in the appendix seems longer ‎than 10 cm.‎

7. PLOS authors have the option to publish the peer review history of their article (what does this mean?). If published, this will include your full peer review and any attached files.

Reviewer #1: No

---

## [Author Response · Author response to Decision Letter 0]

27 Feb 2023

Dear Editor.

The responses for the review process are detailed bellow:

Q: 1. Please ensure that your manuscript meets PLOS ONE's style requirements, including those for file naming. The PLOS ONE style templates can be found at 

R: Updated. Thank you

Q: 2. We note that your submission may include questionnaire items that may have been previously published. The reproduction of previously published work has implications for the copyright that may apply to these publications. We would be grateful if you could clarify whether you have obtained permission from the original copyright holder to republish these items under a CC BY license. If you have not obtained permission to publish these items please remove them from your manuscript. You may wish to replace the text you have removed with relevant question numbers/ brief descriptions of each item; please be sure to include any relevant references and in-text citations.

R: The participants have been included according to the definition of chronic low back pain by different authors. These references have been introduced in each of the questions posed in the questionnaire designed by the research team. Thank you very much for your contribution.

Q: 3. Thank you for stating the following financial disclosure: 

"The funders had and will not have a role in study design, data collection and analysis, decision to publish, or preparation of the manuscript."

R: Line 420. “The authors received no specific funding for this work.” Thank you.

Q: 4. Thank you for stating the following in your Competing Interests section: 

R: Line 421: "The authors have declared that no competing interests exist."

6. Please upload a copy of Figure 3, to which you refer in your text on page 7. If the figure is no longer to be included as part of the submission, please remove all reference to it within the text.

Reviewers' comments:

Reviewer's Responses to Questions

Comments to the Author

Q: 1. Does the manuscript provide a valid rationale for the proposed study, with clearly identified and justified research questions?

R: Reviewer #1: Yes

Q: 2. Is the protocol technically sound and planned in a manner that will lead to a meaningful outcome and allow testing the stated hypotheses?

R: Reviewer #1: Yes

Q: 3. Is the methodology feasible and described in sufficient detail to allow the work to be replicable?

R: Reviewer #1: Yes

Q: 4. Have the authors described where all data underlying the findings will be made available when the study is complete?

R: Reviewer #1: No

Q: 5. Is the manuscript presented in an intelligible fashion and written in standard English?

R: Reviewer #1: Yes

Q: 6. Review Comments to the Author

You may also provide optional suggestions and comments to authors that they might find helpful in planning their study.

R: Reviewer #1: This is a study protocol that investigates the activation time of core muscles via ultrasound in ‎subjects with and without low back pain.‎

Abstract:‎

‎“Observational case-control study." Why did the authors mention “Protocol of an observational ‎cross sectional study” in the title?‎

R:Line 4. Reviewed and changed. Apologies, it was a mistake. Thank you.

Method:‎

Q: Who will diagnose NSLBP? Physician or questionnaire?‎

R: Lines 131 and 754.

A questionnaire will be given to the participant. This questionnaire is designed by the research team and contains information regarding to the inclusion and the exclusion criteria described in the scientific literature. An unblinded subinvestigator (D.V.L and L.L.G) will archive the questionnaires so as not to break the blind.

Thank you

Q: Will the authors consider acute NSLBP or chronic NSLBP?‎ “History of LBP at least three full weeks” for example, a person with 25 days of NSLBP is ‎considered to have acute LBP, while “more than three months in the last year” is a definition for ‎chronic NSLBP.‎

R: Line 144. Only participants with chronic nonspecific low back pain (CNSLBP) defined as twelve weeks last year.

In Oliveira's clinical practice guideline (reference 2), it does provide information on pain duration and classification into acute, subacute and chronic, so this inclusion criterion and the definition of this protocol have been changed according to:

Three guidelines [1, 11, 28] defined acute LBP as less than 4 weeks duration, two guidelines [6, 26] specified less than 6 weeks duration and four guidelines [15, 25, 30, 31] defined acute LBP as less than 12 weeks duration. The Canadian guideline [7] defined acute and subacute LBP as less than 12 weeks duration but without specifying the cutoffs for each one. All guidelines defined chronic LBP as more than 12 weeks’ duration. (Oliveira CB, Maher CG, Pinto RZ, Traeger AC, Lin CWC, Chenot JF, et al. Clinical practice guidelines for the management of non-specific low back pain in primary care: an updated overview. Eur Spine J. noviembre de 2018;27(11):2791-803.

The article: Moissenet F, Rose-Dulcina K, Armand S, Genevay S. A systematic review of movement and muscular activity biomarkers to discriminate non-specific chronic low back pain patients from an asymptomatic population. Sci Rep. diciembre de 2021;11(1):5850. uses this criterion to select the sample for his study, but it is not a definition of the duration of low back pain.

Thank you so much for the correction.

Q: Will the authors include patients with a history of recent participation in core exercises in the ‎past six months? ‎

R: Line 829. We include patients who have participated in core exercises or have practice sports in the last six months as we did not consider it. Upon reading your question, we have updated the questionnaire with a question about sport and frequency. In this way we will try to draw conclusions. Thank you very much.

‎Q: “Balance Mass Index (BMI)”? body or balance?‎

R: Line 379. Body Mass index.

Apologies. Thank you

Q: Did the authors consider the confounding effect of food consumption on the muscle thickness ‎evaluations? (Eur Spine J. 2011 Aug;20(8):1312-7. The effect of food consumption on the ‎thickness of abdominal muscles, employing ultrasound measurements) (Man Ther. 2015 ‎Feb;20(1):194-9. Reversal time of postprandial changes of the thickness of abdominal muscles ‎employing ultrasound measurements).‎

R: The confounding effect of food consumption on the muscle thickness evaluations will not be consider. After a critical reading, the article mentioned is an observational study performed only in men, it is shown that different amounts of food are given for such a heterogeneous sample (very large SD) without considering age, comorbidities, metabolism or even BMI, apparently. They measure, moreover, in a relaxed position and do not give results on the amount of muscle contraction. They do not offer results on how much time would be optimal for food consumption not to affect muscle thickness at rest, nor the amount of food it would be advisable to ingest. They further conclude that ''To minimize this error, it can be suggested that the time between food intake and ultrasound measurements on different days should be consistent for each person''. We think that controlling for this factor is somewhat complicated. The question we would be interested in knowing is whether the contraction-relaxation ratio would be affected by food intake in a short period of time or whether the tension exerted by the stomach affects the functionality of the muscle, even so we insist that the amount of food ingested is key to corroborate these results. 

Thank you very much for your contribution as we had not considered this possible confounding factor.

Q: Will the participants be taught how to perform ADIM before measurements?‎ 

R: Line 238. Yes, the participant will be taught to perform ADIM maneouver before performing the measurement (1.Teyhen DS, Miltenberger CE, Deiters HM, Del Toro YM, Pulliam JN, Childs JD, et al. The use of ultrasound imaging of the abdominal drawing-in maneuver in subjects with low back pain. J Orthop Sports Phys Ther. 2005;35(6):346-55.

Q: Where will the authors assess the internal oblique, external oblique, and transverse abdominis?‎

R: Line 297. Table 2, first line. LAW (IO, EO, TRA) Distance (cm) between the superficial and deep border of each muscle belly, which is marked by the hyperechoic fascial lines. the fascial lines will not be included in the measurement.

Q: Considering the effect of sex, age, and BMI on muscle thickness, why won't the authors adjust ‎their analysis for sex, age, and BMI? (J Physiol Anthropol. 2016 Aug 23;35(1):17. ‎Measurement of superficial and deep abdominal muscle thickness: an ultrasonography study)‎

R: Line 380. We found your contribution very interesting and, after reading the article provided, it fits perfectly with our analysis. Thank you very much.

Q: Is the length of the VAS measurement tool 10 cm? The image in the appendix seems longer ‎than 10 cm.‎ 

R: Yes. It is 10 cm. Thank you

---

## [Editor Report · Decision Letter 1]

24 Apr 2023

Real-time ultrasound evaluation of CORE muscle activity in a simultaneous activation in subjects with non-specific low back pain and without low-back pain. Protocol of an observational cross-sectional study.

PONE-D-22-29823R1

Dear Dr. Fernández Carnero,

We’re pleased to inform you that your manuscript has been judged scientifically suitable for publication and will be formally accepted for publication once it meets all outstanding technical requirements.

Kind regards,

Mehmet Cudi Tuncer, Ph.D.

Academic Editor

PLOS ONE
---

## [Editor Report · Acceptance letter]

2 May 2023

PONE-D-22-29823R1 

Real-time ultrasound evaluation of CORE muscle activity in a simultaneous contraction in subjects with non-specific low back pain and without low-back pain. Protocol of an observational case-control study 

Dear Dr. Fernández Carnero:

I'm pleased to inform you that your manuscript has been deemed suitable for publication in PLOS ONE. Congratulations! Your manuscript is now with our production department. 

Kind regards, 

on behalf of

Professor Mehmet Cudi Tuncer 

Academic Editor

PLOS ONE